# Changes in Expression of Tumor Suppressor Gene RKIP Impact How Cancers Interact with Their Complex Environment

**DOI:** 10.3390/cancers15030958

**Published:** 2023-02-02

**Authors:** Christopher Figy, Anna Guo, Veani Roshale Fernando, Saori Furuta, Fahd Al-Mulla, Kam C. Yeung

**Affiliations:** 1Department of Cell and Cancer Biology, College of Medicine and Life Sciences, University of Toledo, Health Science Campus, Toledo, OH 43614, USA; 2Department of Genetics and Bioinformatics, Dasman Diabetes Institute, Kuwait City 15462, Kuwait

**Keywords:** RKIP, MMP, CCL-5, tumor microenvironment, angiogenesis, ECM, metastasis

## Abstract

**Simple Summary:**

Raf Kinase Inhibitor Protein (RKIP) is a known metastasis suppressor that has regulatory effects on several different pathways. The reduced expression of this protein is strongly correlated with poor prognosis in many cancer types, and several of the cell-specific consequences of its loss have been determined. The tumor microenvironment is the intricate and ever-changing network of extracellular matrix, immune cells, blood and lymphatic vessels, fibroblasts, and signaling molecules that surrounds the tumor. The tumor can facilitate changes to the microenvironment, and the state of the microenvironment can be promotive or restrictive towards tumor growth and metastasis. Loss of RKIP expression has known promotive effects on tumor metastasis and expression of proteins that are able to influence the tumor microenvironment in cancer cells. In this review, we aim to collect and aggregate the relevant research regarding RKIP’s effects on the tumor microenvironment.

**Abstract:**

Tumor microenvironment (TME) is the immediate environment where cancer cells reside in a tumor. It is composed of multiple cell types and extracellular matrix. Microenvironments can be restrictive or conducive to the progression of cancer cells. Initially, microenvironments are suppressive in nature. Stepwise accumulation of mutations in oncogenes and tumor suppressor genes enables cancer cells to acquire the ability to reshape the microenvironment to advance their growth and metastasis. Among the many genetic events, the loss-of-function mutations in tumor suppressor genes play a pivotal role. In this review, we will discuss the changes in TME and the ramifications on metastasis upon altered expression of tumor metastasis suppressor gene RKIP in breast cancer cells.

## 1. Introduction

The tumor microenvironment is constantly evolving during cancer progression due to dynamic interplay between the malignant cells and their microenvironment. In addition to cancer cells, the tumor microenvironment is comprised of, but not limited to, the extracellular matrix, stromal fibroblasts, endothelial cells, immune cells, and secreted factors. It is known that carcinogenesis relies on the accumulation of mutations in oncogenes and tumor suppressor genes. The mutated cancer genome can either directly or indirectly, via the tumor microenvironment, enhance cancer initiation, progression, and metastasis. One of the key tumor suppressor genes whose expression is downregulated as the tumor advances is Raf Kinase Inhibitor Protein (RKIP). The low expression of RKIP is highly correlated with a metastatic phenotype and its level of expression is correlated with clinical outcomes. Importantly, RKIP has a demonstrated causal role in suppressing cancer initiation and metastasis. Initially identified as an inhibitor of the Raf-MEK-Erk signaling pathway, RKIP has since been implicated in regulating numerous different cancer cell signaling pathways. However, the molecular mechanism behind RKIP’s tumor suppressive function remains unclear. Here, we will review the results of previous studies suggesting RKIP may exert its anti-cancer activity through a concerted regulatory control of the tumor microenvironment.

## 2. Loss of RKIP Expression in Cancers

A downregulation or loss of RKIP function is a common event in cancers. However, unlike other tumor suppressor genes, the loss or inactivation of RKIP activity is not a result of RKIP allele deletion or mutational events [1,2]. Few mutations in RKIP alleles were registered on TCGA data banks. Instead, it was suggested that promoter methylation and epigenetic transcription silencing are the two main driving forces for the loss of RKP expression in cancers [1,3].

The cancer genome features an irregular and dramatically altered epigenetic landscape compared to its normal counterparts. Promoter DNA methylation and histone post-translational modifications are common epigenetic mechanisms that control the accessibility of the cancer genome to regulation. Promoter methylation is a major cause of tumor suppressor gene silencing in cancer [2]. It happens when methyl groups are covalently attached to cytosine bases when located in front of guanosines in the vicinity of gene promoters. Promoter methylation causes the shutting down of transcription of the associated gene. RKIP promoter methylation was detected with methylation-specific polymerase chain reaction (MSP) in colorectal cancers, esophageal squamous cell carcinomas, and aggressive gastric cancer [3,4,5]. Significantly, the methylation is associated with reduced RKIP expression, silencing the progression and prognosis of cancers [3,4,5]. However, DNA methylation may not be the sole mechanism to silence RKIP expression in all cancers, as treatment of prostate or liver cancer cells with the methylation inhibitors had no effect on RKIP expression [6].

In addition to promoter methylation, the expression of RKIP in cancer cells can also be silenced by histones’ post-translational modifications. In eukaryotes, DNA is wrapped around histone proteins into compact complexes called nucleosomes. The accessibility of the DNA to transcription regulation is mostly regulated via histone post-translational modifications. Depending on the type of histone modification, a gene can be tuned up or down. Tri-methylation of histone 3 at lysine 27 or 9 (H3-K27-me3 or H3-K9-me3) is usually associated with transcriptional downregulation, di-, or tri-methylation of H3 at lysine 4 (H3-K4-me2/me3); acetylation of H3 at lysine 9 (H3-K9-ace), on the contrary, results in transcriptional activation. Lysine 9 and 27 of histone 3 are extensively tri-methylated in the proximal *RKIP* promoter in low-RKIP-expressing prostate cancer cells. Significantly, the enrichment of the tri-methylated repressive histone modification correlates with the association of a H3 lysine methyltransferase complex, named PRC2, with the proximal RKIP promoter [7].

PRC2 is a multicomponent protein complex with EZH2, Suz12, eed, and RbAp48 as major components. EZH2 and Suz12 are recruited to the proximal RKIP promoter by sequence specific DNA binding transcription factors including Snail and BACH1 [7,8]. Since PRC2 is predominantly a H3 lysine 27 methyltranferase, it is possible that other histone 3 lysine 9 methyltransferases are also involved. Expression of EZH2 is sufficient to inhibit RKIP expression at the transcriptional level. Conversely, the knocking down of the EZH2 expression increases RKIP transcription. Inhibition of invasion due to a loss of EZH2 expression is reversed by the silencing of RKIP. Contrariwise, EZH2-mediated invasion of cancer cells is abolished in the presence of high levels of RKIP. Inhibition of RKIP expression by PRC2 requires its histone lysine methyltransferase catalytic activity. Consistent with its causal role in regulating RKIP expression, the expression of EZH2 is negatively correlated with RKIP expression in cancers [7].

Conceptually, a decrease in RKIP function in cancer can also be a result of an increase in RKIP turnover because of aberrant post-translational regulation. However, evidence supporting the role of post-translational regulation in RKIP expression levels in cancer is lacking. Considering that most of the clinical data linking low RKIP expressions to poor prognosis come from protein expression immunohistochemistry data, post-translation regulation is an area of RKIP that requires pressing attention.

## 3. Observed Changes in TME upon Alternation of RKIP Expression in Cancer

The expression of RKIP diminishes as cancers progress and metastasize [2]. The low expression correlates with a poor prognosis of patient survival in a variety of cancers, raising the possibility that RKIP is a tumor and metastasis suppressor. Indeed, experiments with cancer cells transplantation and genetically engineered mouse models have clearly established the causal role of RKIP in delaying cancer initiation and metastasis [9]. However, the mechanism of how RKIP functions as tumor and metastasis suppressor has not been fully elucidated. Although much of a tumor mass is a heterogeneous population of cancer cells, there are a wide and varied number of resident and infiltrating host cells, secreted factors, and extracellular matrix proteins that constitute the tumor microenvironment. Cancer initiation and progression are complex processes that are greatly dependent on the interactions between cancer cells and the tumor microenvironment. These interactions basically determine whether the primary tumor is annihilated or set for metastasis [10,11]. Therefore, it is possible that RKIP restrains cancer initiation and metastasis by facilitating a more restrictive tumor microenvironment. Truly, changes have been observed in TME when RKIP expression was experimentally altered in breast cancer transplantation mouse models [12,13]. Specifically, it was demonstrated that ectopic expression of RKIP significantly decreases tumor vasculature, macrophage infiltration, and expression levels of MMPs and chemokines in a mouse allograft breast cancer transplantation model. Importantly, these observed changes in the TME were accompanied by a significant decrease in lung metastases. A similar effect of RKIP expression on reducing macrophage infiltration was also observed in primary tumors in a xenograft mouse model with human breast cancer cells [14]. Other than macrophages, the effects of altered RKIP expression on the infiltration of other immune cells (lymphocytes) in the TME have not been investigated extensively. Although no effect on the number of tumor infiltrating B-cells, T-cells, natural killer cells, or neutrophils was observed when RKIP was ectopically expressed in cancer cells [14], it has yet to be established if the observed lack of effect was due to cancer cell type or mouse model used. Considering the pivotal role of tumor infiltrating lymphocytes (TIL) on tumor progression and metastasis, additional studies into the causal relationship between RKIP expression and number of TIL are urgently needed.

## 4. RKIP Inhibits Chemokines Expression to Block Macrophages Infiltration into TME

Tumor associated macrophages (TAM) are derived from monocytes recruited to the TME from the circulation. Upon entry into the TME, these monocytes differentiate into non–activated macrophages (M0) [15]. Later, they polarize into one of several different subtypes of TAMs based on the cytokines and growth factors present within the environment [16]. Additionally, the TAM subtypes present in the TME change with tumor progression. There are two major subtypes present in the TME, which manifest opposing phenotypes. These are the M1-TAMs that possess anti-tumor properties and the M2-TAMs with pro-tumor properties [17]. The microenvironment surrounding the emerging cancer cells is initially restrictive in general, owing to the presence of Th1 type cytokines such as interferon gamma (IFNγ) and granulocyte macrophage colony-stimulating factor (GM-CSF). These cytokines promote the differentiation of monocytes into M1-TAMs. The subsequent secretion of pro–inflammatory cytokines (e.g., TNFα, IL12) and nitric oxide by M1-TAMs forms an auto-regulatory loop that further drives the generation of M1-TAMs. The abundance of M1-TAMs in the TME correlates with a better prognosis of many cancer types [16].

As cancer progresses, tumor cells secrete immunosuppressive Th2 type cytokines, such as IL4, IL10, IL13, and prostaglandin E2 (PGE2). These signaling molecules facilitate M0 polarization into M2-TAMs and are even able to change the phenotype of existing M1-TAMs into M2-TAMs [16]. Once polarized, the M2-TAMs will activate their STAT3 and STAT6 signaling pathways to recruit Th2 (Helper-T cells class2) and Tregs (regulatory T cells) to the milieu. The presence of these cells in the TME confers immunosuppressive activity to the environment. This leads to development of therapeutic resistance in the TME [18,19,20,21]. M2-TAMs also produce growth factors, including insulin like growth factor (IGF), vascular epithelial growth factors (VEGFs), transforming growth factor beta (TGFβ), and matrix metalloproteases (MMPs) which collectively promote tumor growth, angiogenesis, and metastasis [22,23].

Macrophages are noted to play a pivotal role in the development of the tumor microenvironment, and in providing growth and survival signals to the cancer cells. Additionally, the presence of macrophages in the TME has been associated with tumor promotion and stimulation of cancer cell invasion. In the context of the tumor microenvironment, macrophages are also able to secrete angiogenetic growth factors and proteases whose discharge into the surrounding environment has been shown to promote angiogenesis, tumor progression, and metastasis [24,25]. Consistent with its inhibitory role in cancer metastasis, restoring expression of RKIP in low-RKIP-expression breast cancer cells decreased the number of TAMs in vivo [12]. Conversely, the knocking down of RKIP expression increased the TAM number [26]. The effect on TAMs is specific as no significant changes were observed in the number of tumor infiltrating B cells (CD19^+^), T cells (CD3^+^), natural killer cells (CD49^+^), or neutrophils (Gr-1^+^) as assayed by flow cytometry. Besides reducing the number of TAMs, the expression of RKIP also enhances the M2 to M1 macrophage polarization. TAMs isolated from RKIP-expressing tumors displayed a reduced ability to secrete pro-metastatic factors and to drive breast cancer cell invasion in vitro [14]. Mechanistically, forced expression of RKIP reduced the release of extracellular vesicles (EV) from cancer cells, which have been found to be required for M1 to M2 polarization [27].

Cancer cells secrete chemokines to attract monocytes into the TME [28]. RKIP has a negative regulatory role on the expression of multiple chemokines in breast cancer cells [12]. It is possible that RKIP reduces the number of TAMs by interfering with the expression of chemokines. Among them, CCL5 has been directly shown to promote breast cancer by monocyte recruitment to the TME. CCL5 is a pro-inflammatory chemokine. Higher concentrations of the protein have been found in the blood plasma of breast cancer patients, and it is a prominent part of the poor prognosis signature of inflammatory breast cancer. Further analysis via large core needle biopsies of patient samples noted that CCL5 concentrations were significantly higher in patient samples with axillary lymph node metastasis when compared to samples from patients without metastasis [29].

Using ectopic expression of RKIP, it has been shown that RKIP has a significant inhibitory effect on CCL5 levels at the mRNA level. Further, the silencing of RKIP showed the opposite effect. This effect was also confirmed by ELISA at the protein level as well [12]. Loss of RKIP expression increases the proclivity of cancer cells to invade in vitro [30]. It has been shown that the invasive capability conferred to cells upon the loss of RKIP expression is CCL5 dependent. These observations allowed for the generation of a straightforward animal model to confirm the relationship between RKIP, CCL5, and a generally tumor promotive profile. Using an allograft transplantation model with genetically modified 4T1 cells, it was possible to generate tumors and metastases with different expression levels of RKIP and CCL5 in a relevant biological microenvironment. Immunohistochemical staining of these tumors and lung metastases showed a significant decrease in CD31 and F4/80, angiogenesis and macrophage markers, respectively, in samples solely expressing RKIP. The tumors that co-expressed RKIP and CCL5, however, were unable to show this reduction in pro-tumor markers. Further, while RKIP expression is known to suppress metastasis, the co-expressing tumors in this study showed a recovered ability for metastasis, and these metastases showed increased pro-angiogenic signaling [12]. In a separate study, it was reported that the TNBC-secreted CCL5 also stimulated the release of M1 to M2, polarizing the EV via autocrine signaling in addition to recruiting circulating monocytes into the TME [27].

The above findings illustrate the regulatory activity of RKIP towards CCL5, and the importance this relationship has in cancer prognosis. There is a known inverse correlation between RKIP and CCL5 in breast cancer datasets, in addition to there being direct correlations between loss of RKIP and poor prognosis, and heightened CCL5 expression and poor prognosis. Further, the relationship between CCL5 expression levels and metastasis has been illustrated, where the majority of metastasis associated patient samples have high CCL5 expression in their primary tumors [12]. This suppressive relationship RKIP has towards CCL5, one of the more potent pro-tumor chemokines, makes RKIP a prime target for silencing by the tumor in its advancement towards metastatic disease.

The mechanism of how RKIP inhibits the transcription of CCL5 is not well defined. Of note, the promoter region of the CCL5 gene possesses both NF-kB and IRF transcription factors’ relevant binding sites [31]. Importantly, co-expression of IRF-3, IRF-7, and NF-kB was shown to be needed for synergistic activation of CCL5 promoter transcription [31]. The requirement for normal NF-kB and IRF signaling for the observed heightened CCL5 expression in breast cancer tumors has been confirmed. In view of the demonstrated regulatory role of RKIP on IRFs and NF-kB activities [32,33], it is possible that RKIP affects CCL5 transcription by targeting these transcription factors. In addition to targeting NF-kB and IRF, RKIP inhibits architectural transcription factor high-mobility group AT-hook 2 (HMGA2) indirectly through the ERK-myc-lin28-let7 pathway [34]. It was shown that the knocking down of HMGA2 by specific siRNAs significantly decreased *CCL5* transcripts in cultured breast cancer cells [14]. HMGA2 does not regulate gene expression directly, but indirectly by altering and bending the DNA structure to promote the assembly of transcriptional protein complexes that regulate transcription initiation. It remains to be determined if HMGA2 is needed for the cooperative synergism between IRF-3/IRF-7 and NF-kB factors required for optimal CCL5 expression. Furthermore, in a recent study with a 4T1 allograft transplantation model, the small GTPase RhoA was shown to function downstream of RKIP and upstream of CCL5 as suppressor of breast cancer lung metastasis [26]. The mechanism of how RhoA regulates CCL5 expression is presently unknown.

## 5. MMP and ECM Remodeling

Extracellular matrix is the acellular component of the TMC, comprising a complicated network of macromolecules. Its specific composition is usually tissue dependent, but some common components such as collagen and proteoglycans, such as laminin, and fibronectin, can be considered ubiquitous for their presence in the ECM. These macromolecules, secreted mainly by stromal fibroblasts, assemble into a vast three-dimensional scaffold to provide normal resident cells with architectural and mechanical support and protection [35]. The ECM is also a storage depot of a diverse pool of growth factors and cytokines, providing cues when released to regulate and fine-tune cellular homeostasis, and determine cell fate. The structure and components of the matrix are plastic and dynamic, and subjected to breaking down and rebuilding in response to tissue damage, or during development. One of the important mediators of the ECM restructuring are the matrix metalloproteinases (MMPs) [36,37].

Matrix metalloproteinases (MMPs) are a family of calcium-dependent zinc-endopeptidases that cleave most, if not all, components of the ECM. The family contains over 20 members of secreted and membrane bound endopeptidases, and has a large range of enzymatic targets in addition to components of the ECM. Based on their specificity toward ECM components, MMPs can be divided into collagenases, gelatinases, stromelysins, and matrilysins. According to their structural motifs, MMPs can also be grouped into eight distinct groups with five secreted types, and the remaining are membrane-type MMPs (MT-MMPs). MMPs have classically been defined by their homologous ‘minimal domain’, which contains the pro-peptide and the zinc-binding associated catalytic domain. The MMPs pro-peptide contains the PRCGXPD “cysteine switch” motif. Cysteine from the “cysteine switch” motif interacts with the zinc ion at the catalytic site of the enzyme, which blocks its activity. The pro-peptide interaction with the zinc ion needs to be interrupted for its catalytic activity to be available; thus, the pro-peptide domain contains a consensus sequence and requires proteolytic cleavage by convertases, which, depending on the sequences, occurs intracellularly by furin or extracellularly by other MMPs or serine proteinases such as plasmin [38]. The catalytic domain differentiates between different MMPs in specific enzymatic activity.

The ECM in normal tissues is hostile to burgeoning cancer cells. Collagen-derived peptides such as canstatin, tumstatin, or tetrastatin can restrict tumor growth by induction of apoptosis and inhibition of the migratory phenotype in tumor cells [39,40]. Basement membranes that separate the epithelial cells from the underlying extracellular matrix are specialized ECMs. The interconnected and compact networks of type IV collagen and proteoglycans in the basement membrane present a structural barrier to invading cancer cells [41]. In addition, the oxygen, glucose, and essential metabolite content of the normal cellular microenvironment fails to meet the demand of frantically dividing emerging cancer cells. However, the stepwise accumulation of mutations in oncogenes and tumor suppressor genes confers upon a few cancer cells the ability to overcome the barriers to growth imposed by its microenvironment. This includes the ability to remodel the ECM and increase its vasculature by stimulating angiogenesis [42]. Collagen reorganization and modification of proteoglycan expression in the TME facilitate cell invasion and metastasis [43]. Tumors are generally noted to have increased stiffness due to several factors. Cancer cells can coax infiltrating fibroblasts to increase their ECM deposition, and the increased presence of lysyl oxidase (LOX) and transglutaminase in the TME results in increased cross-linking between collagen and elastin [37]. These larger collagen and elastin fibrils, compared to adjacent tissue, have been observed to help facilitate cancer cell migration [43]. It has been shown that the degradation of the ECM macromolecules can release a variety of growth factors and angiogenic factors [44]. One of the decisive hallmarks in cancer progression has been observed to be the disruption of the architecture of the basement membrane. This specialized ECM is more compact and less porous, and generally provides the supporting structure which surrounds resident epithelial and endothelial cells. It acts as a physical barrier to metastatic cell intravasation into the circulation, and proteolytic degradation of the basement membrane is an important prognostic marker for invasive cancer [45].

Given their ability to degrade the ECM, including the basement membrane, and release entrapped growth and angiogenic factors, MMPs expression levels are frequently upregulated in cancers [38]. One of the mechanisms that cancer cells can exploit to increase MMP activity is the inactivation of the tumor suppressor genes that are important for MMPs gene expression. RKIP is negatively correlated with the expression levels of several MMPs in several cancer types [30]. Further investigation into this relationship has shown that the expression signature of RKIP and these MMPs together can better predict high metastatic risk than analysis of either gene alone [13]. Using an in vitro cell-based system, it was demonstrated that RKIP inhibits the transcription of several MMP transcripts including MMP1-3, MMP10, and MMP13 in cancer cells [13]. Of note, all RKIP-regulated MMPs are secreted types. While MMP1 and MMP13 are collagenases, MMP2 is a gelatinase, and MMP-3 and MMP-10 are stromelysins. Apparently, RKIP targets secreted MMPs that digest different components of the ECM. At present, it is not known if RKIP influences the processing of inactive MMPs into active MMPs. It is known that RKIP’s metastatic suppressive capabilities are at least in part due to suppression of MMP1, MMP2, and MMP13 expression, as a loss of RKIP results in increased MMP expression and increased invasiveness [30] (Figure 1A).

It is apparent that RKIP suppresses MMP expression through a variety of different pathways, including the Raf/MEK/ERK pathway and the NF-kB pathway (Figure 1A). There is also evidence to suggest that RKIP negatively regulates MMP expression through the transcriptional repressor Snail and activator BACH1 [30,34]. Given the combination of different regulatory paths that RKIP engages with to negatively influence the expression of a variety of MMPs, it is likely that targeted suppression of RKIP expression in cancers plays a major role in facilitating the cancer’s ability to upregulate MMP expression and further modulate its TME to produce favorable and proliferative conditions.

The requirement of ERK2 for the relationships between RKIP and MMP13 expression has been confirmed, indicating a general regulatory flow of RKIP to ERK2 to MMP13 through a currently unknown transcription factor [13] (Figure 1A). Although Raf/MEK/ERK-mediated regulation of transcription factors is involved in the regulation of MMP13 by RKIP, it has been determined that this pathway is not required for the observed invasive nature of cancer cells with regards to RKIP and MMP1 and MMP2 in vitro [30]. The regulatory role of RKIP in MMP1 and MMP2 expression appears to be through RKIP’s inhibitory regulation of the NF-kB pathway, as inhibition of this pathway nearly fully inhibits the increased invasive capabilities observed in cultured cancer cells when RKIP is lost [30] (Figure 1A).

RKIP inhibits the expression of multiple MMPs in cancer cells [13]. Although the decrease in MMPs expression is the cause of RKIP-mediated suppression of cancer cell invasion in vitro, the effects of MMPs on the metastasis cascade in vivo are MMP specific [13,30,46]. A significant decrease in the number of cancer cells found in tumor draining lymph nodes, circulating cancer cells, and lung metastases were observed when RKIP was expressed in an orthotopic mouse breast cancer model compared to controls. The detection of cancer cells in the lymph nodes can be restored to control levels upon re-expression of MMP13 in these RKIP expressing cells (Figure 1B); however, the effects of RKIP on extravasation and lung metastasis are MMP13 independent, as re-expression of MMP13 failed to reverse the effect due to the expression of RKIP. In a different system where breast cancer cells were directly injected into the cardiac ventricle of immunocompromised mice to score for bone metastasis, expression of RKIP significantly decreased metastasis. Co-expression of MMP1 and OPN (osteopotin) is sufficient to rescue the effect on bone metastasis due to the expression of RKIP. Using the same system, RKIP was shown as negatively regulating the expression of MMP1 through the ERK-myc-lin28-let7-BACH1 transcriptional axis in vivo (Figure 1B). Since RKIP was also shown to regulate MMP1 expression through the NF-kB pathway in vitro, it suggests that a different pathway is apparently targeted by RKIP to regulate MMP1 expression in vivo [46]. Taken together, this evidence indicates that RKIP targets specific MMPs to inhibit different steps of the metastasis cascade, and may use different mechanisms to regulate the expression of the same MMP in different contexts.

## 6. Angiogenesis

RKIP has been established as a cancer metastasis suppressor in cancer cell transplantation models [47]. It has been hypothesized that one mechanism by which RKIP suppresses metastasis is through suppressing angiogenesis within primary tumor tissue. Indeed, when RKIP expression is reduced in prostate tumors, the density of CD31-positive microvasculature is significantly increased when compared to tumors which expressed wildtype levels of RKIP [48]. These findings point toward RKIP having some anti-angiogenic properties. The process of angiogenesis in cancer is regulated at multiple levels and is heavily influenced by the TME, as well as the stage of the cancer [49]. There exist several mechanisms by which RKIP has the potential to modulate this process.

Tumor cells rely heavily on glucose, oxygen, and essential metabolites to be supplied by existing blood vessels to sustain rapid mitotic activities. To support this elevated metabolic demand, tumors are hotbeds for the generation of new vascular vessels through angiogenesis. It is a multi-step process that involves the dissolution of the basement membrane and ECM around existing blood vessels, followed by the proliferation and re-positioning of endothelial cells, and finally the transformation into capillary tubes [50]. This process involves intricate crosstalk between members of the TME, including tumor-associated macrophages (TAMs), cancer-associated fibroblasts (CAFs), extracellular matrix components, and the cancer cells [51,52]. Angiogenesis is regulated by several pro- and anti-angiogenic factors that are released by the cancer or stroma cells, resulting in a balance between the two. Initially, this balance is anti-angiogenic to maintain quiescence of the established vasculature. Cancer cells shift this balance in favor of pro-angiogenic signaling in a process known as angiogenic switch [53]. Pro-angiogenetic factors include various growth factors such as vascular endothelial growth factor (VEGF), basic fibroblast growth factor (bFGF), and angiopoietin. While positive angiogenic regulators are mostly growth factors, anti-angiogenic factors can be cytokines or peptides such as endostatin and tumstatin derived from collagen in the ECM [49].

Cancers achieve the angiogenic switch by different methods. As the tumor grows and rapidly proliferates, the surrounding environment will often suffer from a loss of oxygen. This reduction in oxygen in the surrounding environment generates intratumoral hypoxic conditions, sometimes called the hypoxic niche, and represents one of the many factors influencing the angiogenic switch [44]. These hypoxic conditions stimulate the activity and stabilization of the HIF-1a subunit of the transcription factor HIF-1 (hypoxia-inducing factor-1) in both cancer and stromal cells [54]. In addition, HIF activity is also known to be inducible in an oxygen-independent manner. Specifically, the activation of the Raf-MEK-Erk or PI3K/Akt signaling pathways results in increased HIF-1 activity by upregulating the HIF-1a protein translation in an mTOR independent or dependent manner, respectively [54,55]. HIF targets many genes to regulate a diverse array of biological pathways. Active and stabilized HIF targets genes that are involved in angiogenesis, glucose metabolism, cancer cell survival, and proliferation. In light of the findings that RKIP negatively regulates both Raf-MEK-Erk and PI3K/Akt signaling pathways [56,57], the ability of RKIP to modulate angiogenesis by regulating HIF-1a activity has been considered.

VEGF (vascular endothelial growth factor) is one of the key and well-studied pro-angiogenic factors associated with tumor angiogenesis. VEGF is produced by cancer cells and many of the stromal TME components, and it binds to and activates vascular endothelial cells. This induces an increase in vascular permeability, proliferation, and migration of these cells to promote angiogenesis. Cancer cells employ several mechanisms to influence the amount of VEGF present in the TME. In response to hypoxia, tumor cells stimulate transcription factor HIF-1a production, which upregulates the synthesis of VEGF. Moreover, HIF-1 also increases glycolysis and the release of lactate acid into the TME, and lactic acid induces tumor-associated macrophages to secrete more VEGFA [49,51].

The acidification of the microenvironment also disrupts the activation of transcription factor NFAT (Nuclear factor of activated T cells), which is required for the transcription of cytokine genes such as interferon g in NK and T-cells, and the inhibition of cytotoxic T-cell activity. In addition, cancer cells can activate fibroblasts to metabolize tryptophan, an essential amino acid for immune cells, and produce the immunosuppressive metabolite kynurenine. It is proven that kynurenine inhibits cell proliferation and incites apoptosis in T-cells. Chronic VEGFA signaling promotes glycolysis in endothelial cells and feeds into this positive feedback loop: tumor growth generates a hypoxia-induced acidic and nutrient-starved tumor microenvironment, which then promotes further tumor growth through angiogenesis and immunosuppression [58].

Although RKIP may possibly interfere with VEGF expression by inhibiting HIF-1a translation via Raf or PI3K pathways, modulation of RKIP expression has no significant effect on the expression of VEGF in cancer cells [48]. These findings, therefore, indicate that the anti-angiogenic influence of RKIP is not through the inhibition of VEGF expression in cancer cells. As noted previously in this review, RKIP’s regulatory role on the expression of CCL5 in cancer cells influences the recruitment of macrophages to the TME. Tumor-associated macrophages (TAMs) are major producers of pro-angiogenic factors, representing a key group of cells responsible for the angiogenic switch. As such, suppression of TAM recruitment by RKIP through CCL5 represents a likely mechanism by which RKIP regulates angiogenesis. Indeed, inducing CCL5 expression is able to rescue the anti-angiogenic effects caused by RKIP expression, suggesting that the regulatory role of RKIP in tumor angiogenesis involves indirect pathways [12].

RKIP expression is significantly correlated to the expression levels of several different MMPs, including MMP13 [30]. Several studies by others have highlighted the importance of these enzymes in the angiogenic processes of cancers. For example, the observed vascularization in transplanted tumors isolated from mice with germline knockout of MMP13 was significantly reduced compared to wild-type mice [59]. As expected, the tumors in mice without the expression of stromal MMP13 were less invasive. Mechanistically, MMPs regulate angiogenesis primarily through demolishing ECM components and digesting away the basal lamina of endothelial cells. This results in the release of various entrapped angiogenic factors and allows endothelial cells to invade the tumor stroma forming new blood vessels [37,50]. Interstitial collagens are the major components in the ECM. The triple-helical, fibrillar structure in collagens is highly resistant to many proteases including many MMP members. Among the few MMPs that can cleave the highly structured fibrillar collagens, three of them, namely MMP1-2 and MMP13, are regulated by RKIP [30]. These are the MMPs that can digest well-structured fibrillar collagens in the ECM around the blood vessel, and add to the release of ECM-bound angiogenic regulators. Through regulation of the expression of these proteins, RKIP has additional pathways by which it can regulate angiogenesis and inhibit cancer cell invasion.

## 7. Conclusions

The malignant transformation of a cell is initiated by autonomous oncogenic events, which directly determine the cancer genome and also indirectly shape its immediate environment (tumor microenvironment) to generate non-autonomous conditions favorable for tumor growth. There are multiple lines of emerging evidence to suggest that the tumor and metastasis suppressor RKIP functions by interfering with the promotive interaction of cancer cells with their environment. It has been shown that the presence of RKIP suppresses multiple different mechanisms that the tumor would use to promote or communicate with the TME to generate promotive effects (Figure 2). As the tumor progresses in severity, there is generally an associated loss of RKIP expression which is further associated with an increased expression of different TME-altering pathways. In this review, we have shown a causal link between RKIP and several of these TME mediating proteins and illustrated how the reduction in RKIP expression, which is common in many tumor types, results in an increase in the activity of these related processes. The ability of RKIP to suppress these different tumor-promoting effects, therefore, provides a strong rationale for restoring RKIP expression by identifying therapeutic targets that silence RKIP in cancer.

## Figures and Tables

**Figure 1 cancers-15-00958-f001:**
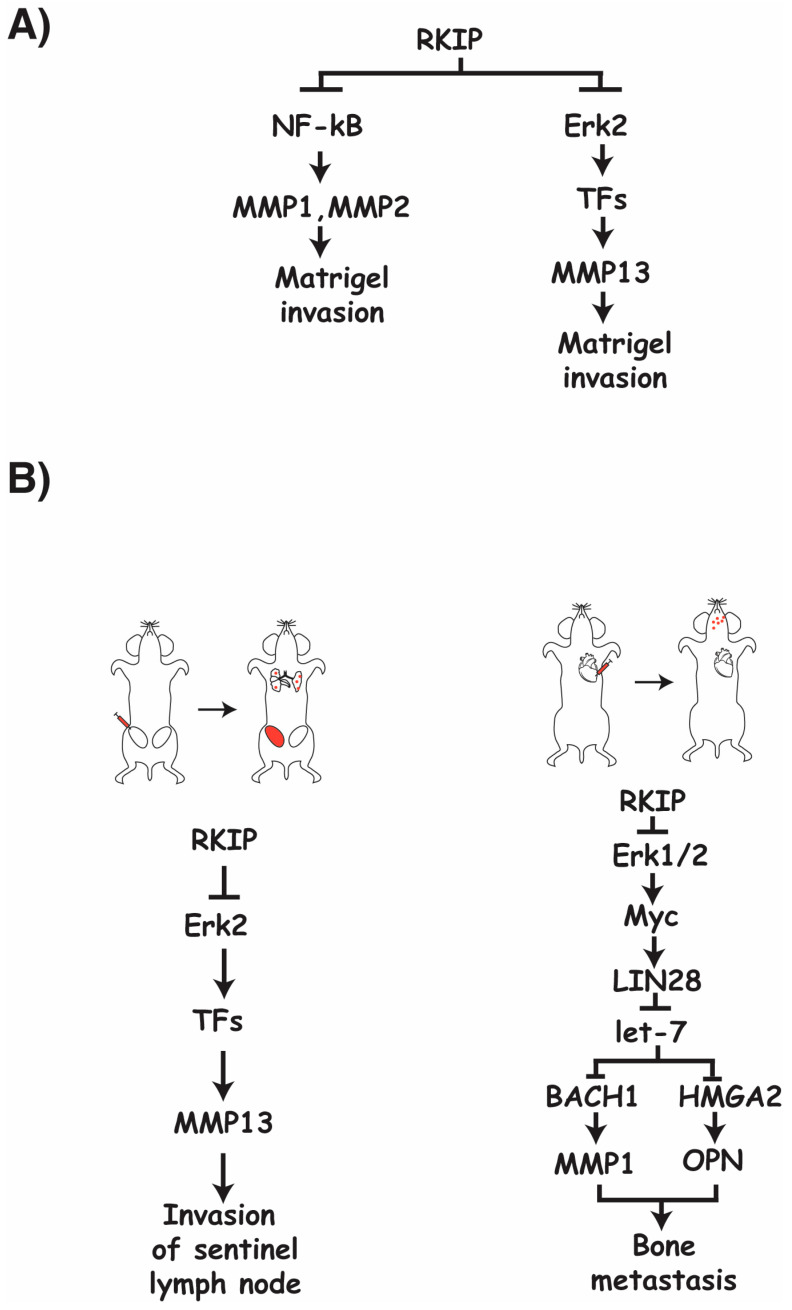
(**A**) A schematic summary of signaling pathways that are targeted by RKIP to regulate the indicated MMP expression in vitro. (**B**) Cartoons showing two different cancer transplantation models used to identify signaling pathways targeted by RKIP to regulate MMP expression in vivo.

**Figure 2 cancers-15-00958-f002:**
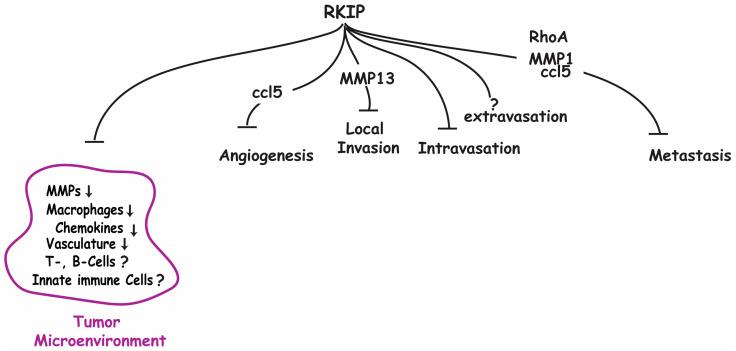
A schematic summary of the causal role of RKIP on the expression and effects of several proteins known to modulate the TME, and the metastasis cascade.

## Data Availability

Not applicable.

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
