# Peer review of "Changes in Expression of Tumor Suppressor Gene RKIP Impact How Cancers Interact with Their Complex Environment"

_cancers, 2023, doi:10.3390/cancers15030958_

Round 1

Reviewer 1 Report

The present work of Figy et al. describes the role of RKIP in regulating components of the tumor microenvironment. This is an interesting work that can help to clarify some of  the mechanisms by which RKIP intervenes in the progression of tumors. Howevere, In order to make this work even more complete, it would be useful to investigate also the relationship between RKIP and the activation/inhibition of T lymphocytes that is only marginally described. 

In fact, tumor infiltrating lymphocytes (TIL) play a key role in combating tumor progression and, vice versa, their inactivation contributes to one of the mechanisms underlying the spread and metastization. 

Since RKIP blocks tumour progression, the role between dysregulation of RKIP exposure and TIL activation/inactivation should be described.

In addition, it would be appropriate to include at least a figure and a table to simplify the understanding of the many interactions between RKIP and the various components of TME.

Author Response

Response to reviewer' comments:

Original comments are in italics. We also marked the changes in the re-submitted manuscript with underlines.

The present work of Figy et al. describes the role of RKIP in regulating components of the tumor microenvironment. This is an interesting work that can help to clarify some of the mechanisms by which RKIP intervenes in the progression of tumors. However, in order to make this work even more complete, it would be useful to investigate also the relationship between RKIP and the activation/inhibition of T lymphocytes that is only marginally described. 

In fact, tumor infiltrating lymphocytes (TIL) play a key role in combating tumor progression and, vice versa, their inactivation contributes to one of the mechanisms underlying the spread and metastization. 

Since RKIP blocks tumour progression, the role between dysregulation of RKIP exposure and TIL activation/inactivation should be described.

A paragraph that summarizes published results on the relationship between RKIP expression and number of TIL has been added.

In addition, it would be appropriate to include at least a figure and a table to simplify the understanding of the many interactions between RKIP and the various components of TME.

Thanks for the suggestion. Two figures have been added to enhance the clarity of the review.

Reviewer 2 Report

1. The authors discuss evidence of histone modifications and promoter methylation as mechanisms of RKIP silencing. Are there studies investigating post-translational regulation of RKIP expression such as ubiquitination or any other regulation that affects protein turnover? This is important because the majority of the clinical data that connect low RKIP expression to worse clinical outcomes comes from immunohistochemistry data which shows changes in RKIP protein levels. This is not always correlated with changes in RNA levels. The authors should discuss the evidence regarding post-translational regulation of RKIP as well, and if there is not enough evidence towards this, then the authors should highlight it as an area of RKIP research that requires further investigation.   

2. A recent study published in MDPI-Cancers [1] suggests that RKIP-CCL5 axis might not only be important for the recruitment of macrophages, but it can also reprogram macrophages along the M1-M2 axis through extracellular vesicles. This study can be included in the review as it provides a mechanistic link between RKIP-CCL5 and M2 macrophages.

3. The detailed description of MMP types in lines 221-237 is not necessary as these specific subtypes of MMP are not discussed in the context of RKIP in the manuscript, and frankly, the description can be confusing to some readers. If the authors choose to keep this paragraph as is, they might want to include a simple figure depicting the different MMPs described.

4. This manuscript is begging for a summary illustration of all the different mechanisms by which RKIP remodels TME. The manuscript would greatly benefit from a figure like this.

5. Line 17 – expression of proteins “that are” able to influence the tumor microenvironment…

6. In general a study should be cited at the end of the first sentence a finding from that study is being discussed. The reader assumes that the following sentences are referring to that study. The authors seem to cite their studies after having discussed all the findings from that study within a paragraph, which makes the earlier statements in the paragraph look like unsupported evidence. For example, in Lines 108-112 and Lines 294-297, these statements should be cited right away, not later in the paragraph.

7. Line 157 – define “TMC” for the first time you use the abbreviation

8. Overall, the manuscript will benefit greatly from a thorough grammar check.

References

[1] Rabe, D.C.; Walker, N.D.; Rustandy, F.D.; Wallace, J.; Lee, J.; Stott, S.L.; Rosner, M.R. Tumor Extracellular Vesicles Regulate Macrophage-Driven Metastasis through CCL5. Cancers 2021, 13, 3459. https://doi.org/10.3390/cancers13143459

Author Response

Response to reviewer comments:

Original comments are in italics. We also marked the changes in the re-submitted manuscript with underlines.

  1. The authors discuss evidence of histone modifications and promoter methylation as mechanisms of RKIP silencing. Are there studies investigating post-translational regulation of RKIP expression such as ubiquitination or any other regulation that affects protein turnover?

We cannot find any published study investigating post-translational regulation of RKIP expression that affects protein turnover in Pubmed. A statement has been added to the revised manuscript.

This is important because the majority of the clinical data that connect low RKIP expression to worse clinical outcomes comes from immunohistochemistry data which shows changes in RKIP protein levels. This is not always correlated with changes in RNA levels. The authors should discuss the evidence regarding post-translational regulation of RKIP as well, and if there is not enough evidence towards this, then the authors should highlight it as an area of RKIP research that requires further investigation.

We agree. A statement has been added to the revised manuscript underscoring the importance of this area of RKIP research.

  1. A recent study published in MDPI-Cancers suggests that RKIP-CCL5 axis might not only be important for the recruitment of macrophages, but it can also reprogram macrophages along the M1-M2 axis through extracellular vesicles. This study can be included in the review as it provides a mechanistic link between RKIP-CCL5 and M2 macrophages.

Thanks for the information. A paragraph that includes the mentioned study has been added to the review.

3a. The detailed description of MMP types in lines 221-237 is not necessary as these specific subtypes of MMP are not discussed in the context of RKIP in the manuscript, …

A paragraph that describes different MMP subtypes in the context of RKIP has been added to the review.

3b. …and frankly, the description can be confusing to some readers. If the authors choose to keep this paragraph as is, they might want to include a simple figure depicting the different MMPs described.

Thanks for the suggestion. A figure has been added for the clarity.

  1. This manuscript is begging for a summary illustration of all the different mechanisms by which RKIP remodels TME. The manuscript would greatly benefit from a figure like this.

Thanks for the suggestion. A summary figure has been added to the review.

  1. Line 17 – expression of proteins “that are” able to influence the tumor microenvironment…

Change has been made. Thank you.

  1. In general a study should be cited at the end of the first sentence a finding from that study is being discussed. The reader assumes that the following sentences are referring to that study. The authors seem to cite their studies after having discussed all the findings from that study within a paragraph, which makes the earlier statements in the paragraph look like unsupported evidence. For example, in Lines 108-112 and Lines 294-297, these statements should be cited right away, not later in the paragraph.

Changes have been made. Thank you.

  1. Line 157 – define “TMC” for the first time you use the abbreviation

Change has been made. Thank you.

  1. Overall, the manuscript will benefit greatly from a thorough grammar check.

A thorough grammar check has been performed. Thank you. 

Round 2

Reviewer 1 Report

The reviewed version of the MS has been appropriately  improved and can be accepted in the present form.